

# Validation of reanalysis Southern Ocean atmosphere trends using sea ice data

William R. Hobbs[1,2], Andrew R. Klekociuk[1,3], Yuhang Pan[4]

[1]Australian Antarctic Program Partnership, Institute of Marine and Antarctic Studies, Private Bag 129, University of Tasmania,
Hobart, TAS 7001, Australia.
[2]ARC Centre of Excellence for Climate Extremes, Institute for Marine and Antarctic Studies, University of Tasmania, Hobart,
TAS 7001, Australia.
[3]Antarctica and the Global System Program, Australian Antarctic Division, 203 Channel Highway, Kingston, TAS 7050,
Australia.
[4]Independent Researcher

*Correspondence to*: William R. Hobbs (will.hobbs@utas.edu.au)

**Abstract.** Reanalysis products are an invaluable tool for representing variability and long-term trends in regions with limited
in-situ data, and especially the Antarctic. A comparison of 8 different reanalysis products shows large differences sea level
pressure and surface air temperature trends over the high latitude Southern Ocean, with implications for studies of the
atmosphere's role in driving ocean-sea ice changes. In this study, we use the established close coupling between sea ice cover
and surface temperature to evaluate these reanalysis trends using the independent, 30-year sea ice record from 1980-2010. We
demonstrate that sea ice trends are a reliable validation tool for most months of the year, although the sea ice-surface
temperature coupling is weakest in summer when the surface energy budget is dominated by atmosphere-to-ocean heat fluxes.
Based on our analysis, we find that surface air temperature trends in JRA55 are most consistent with satellite-observed sea ice
trends over the polar waters of the Southern Ocean.

## 1 Introduction

Atmospheric trends in the Southern high latitudes have global importance. Wind patterns are essential for driving the Southern
Ocean overturning, which is responsible for most of the global ocean's uptake of anthropogenic heating, and approximately
half its uptake of anthropogenic carbon (Frolicher *et al*, 2015). Local wind changes are a factor in the ocean melting of West
Antarctic ice shelves (Lenaerts *et al*, 2017; Paolo *et al*, 2018; Dotto *et al*, 2019), with implications for global barystatic sea
level rise (Dupont and Alley, 2005; Pritchard *et al*, 2012; Paolo *et al*, 2015), and polar winds are clearly related to
observed Antarctic sea ice trends (Holland and Kwok, 2012). More immediately, variability in the Southern Annular Mode –
the dominant mode of mid-high latitude Southern Hemisphere atmospheric variability – is thought to influence Australian
rainfall (Meneghini *et al*, 2007), with implications for current and future droughts. Clearly, a reliable and accurate
representation of high latitude Southern Hemisphere atmosphere trends is essential.

For this data-sparse region, atmospheric reanalysis products are the primary research tool for analyzing observed changes or as
surface boundary conditions for ocean-sea ice models. However, there is a wide spread in surface atmosphere trends over the



Southern Ocean amongst different reanalysis products, which introduces uncertainty when interpreting observed ocean and
        sea ice trends (Marshall, 2003; Swart and Fyfe, 2012; Hobbs *et al*, 2016). Reanalysis validation studies have attempted to
        address this uncertainty, but have largely been restricted to comparisons with long-term surface measurement sites, almost all
        of which are located near the Antarctic coast (e.g. Turner et al., 2014); relatively few studies have been conducted for the sea-
        ice zone (Bracegirdle and Marshall, 2012; Jones *et al*, 2016).

Figure 1 shows linear trends in 2m air temperature (SAT) and mean sea level pressure (MSLP) from eight commonly-
        used atmosphere reanalyses (summarized in Table 1), and clearly demonstrates this spread in trends. Some reanalyses show
        almost no warming at all in West Antarctica, whilst NCEP2 shows a warming over the entire sea ice zone. Station data show
        a   distinct asymmetry in   the   long-term   behavior  of SAT between Antarctica's eastern   and   western   hemispheres,
        with the statistical significance of trends depending on epoch, season and location. A general warming has occurred in recent

decades in the Antarctic Peninsula and parts of West Antarctica, with weaker mixed trends in East Antarctica (Marshall *et al*,
        2013;   Nicolas   and   Bromwich,   2014;   Turner *et   al*,   2014). Compared   with station measurements, SAT trends
        in reanalyses show less consistency with spurious behavior in some regions, particularly in East Antarctica where surface
        stations are sparse (Bromwich *et al*, 2013; Steig and Orsi, 2013; Wang *et al*, 2016; Simmons *et al*, 2017). However, there is
        generally   better   agreement between  observations  and reanalyses when  interannual  variability  rather  than  trends  are

considered (e.g. Wang *et al*, 2016).

        There is a similar spread in MSLP trends; many of the reanalyses show the widely-reported deepening of the Amundsen Sea
        Low (Hosking *et al*, 2013; Turner *et al*, 2013; Raphael *et al*, 2016) – although with some disagreement on magnitude and exact
        location – but by no means all of them. Additionally, there is a known spread amongst reanalyses in the magnitude of the
        Southern Annular Mode positive trend (Marshall, 2003; Swart and Fyfe, 2012). This raises the question of which

representation is the most accurate, for interpreting recent historical changes in the atmosphere-ocean-cryosphere system of
        the polar Southern Ocean.

        There is a close link between sea ice cover and the atmosphere, both for interannual variability and at longer time scales (e.g.
        Comiso *et al*, 2017). Atmospheric thermal advection modulates the rate of sea ice freeze/melt, and wind driven ice motion
        redistributes the existing sea ice. In the Southern Ocean the sea ice-atmosphere relationship tends to be stronger in the sea ice

growth season and weaker in the melt season (Raphael and Hobbs, 2014; Schroeter *et al*, 2017). This may be because
        approximately half of the heat driving sea melt comes from the ocean (Gordon, 1981), diminishing the relative impact of the
        atmosphere. The atmosphere-sea ice relationship is particular strong for surface air temperature (e.g. Comiso *et al*, 2017), due
        to positive feedbacks: a colder air temperature leads to increased sea ice cover, which due to increased albedo and much
        reduced ocean-to-atmosphere heat flux can further reduce air temperature. In short, sea ice affects air temperature, and air

temperature affects sea ice.

        Previous studies have exploited this close relationship to study sea ice. Notably, King and Harangozo (1998) demonstrated a
        close link between Antarctic Peninsula station temperature and local sea ice changes, Massonnet *et al* (2013) were able to
        reproduce Antarctic sea ice variability in a model driven by SAT, and both Kusahara *et al* (2017) and Schroeter *et*



*al* (2018) showed the important role that thermodynamic forcing has on Antarctic sea ice trends. This close coupling between

SAT and sea ice concentration (SIC) indicates that the passive microwave sea ice record may be used as an independent validation of Reanalysis SAT trends, at least for the broad spatial patterns that are clearly different in Figure 1.

In this study, we perform just such an evaluation. We demonstrate that SIC and SAT variability are closely related for much of the year, except for the season of strongest sea ice melt. Based on that premise, we find that a number of reanalysis products have trends that are physically-consistent with independently-observed sea ice trends, with ERA5 showing a marginally-better

agreement than other products. A smaller group of products are very obviously inconsistent with the sea ice trends, and should be avoided for studies of long-term change in the high-latitude Southern Hemisphere. We argue that the weak SAT-SIC relationship in summer is due to the direction of ocean-atmosphere heat flux in those months; since the net balance in the sea ice zone is *from* atmosphere *to* ocean, the surface energy budget is more a response to - rather than a driver of – the near-surface atmosphere.


**2 Data and Method**

We use monthly mean SIC from passive microwave satellite observations as the primary dataset for evaluating reanalysis SAT trends. Specifically, we use the Goddard-merged data from the NOAA/NSIDC climate data record for SIC, available on a 25 km x 25 km equal area grid (Meier *et al*, 2014).

We analyze monthly-mean SIC, SAT and MSLP from 8 publicly-available reanalysis products, which are summarized in Table 1. These products span a range of spatial resolutions, assimilation algorithms and analysis periods. For this study, we consider the period 1980-2010 inclusive, which is the longest period covered by all 8 reanalyses, constrained by MERRA2 (starting in January 1980) and ERA-20C (ending in December 2010), and matches the period of the SPARC Reanalysis Intercomparison Project (S-RIP: Fujiwara *et al*, 2017).

Although we consider the relationship between SAT and SIC at interannual timescales, our primary focus is on the 31-year trends of the analysis period, calculated by month using Ordinary Least Squares regression. To quantify the level of agreement between trend patterns for SIC and reanalysis SAT, we use an uncentered pattern correlation (i.e. without removing spatial means), applying a cosine-weighting to account for latitude-dependence of the grid area. To facilitate this, all variables were regridded onto a common 1° x 1° latitude/longitude grid using bilinear interpolation.


**3 Results**

**3.1 Evaluation of SAT based on sea ice trends**

Although sea ice trends are not themselves the focus of this work, except as an independent validation of reanalysis SAT, the observed SIC trends are shown in Figure 2 for illustrative purposes. (Note that while the trends are aggregated in Figure 2 into

seasons defined by sea ice melt/growth, for the SAT validation we used monthly trends, shown in Supplemental Material). The trend patterns are well-established and have been described in many previous studies (e.g. Parkinson and Cavalieri, 2012;



Hobbs *et al*, 2016; Comiso *et al*, 2017), and can be broadly summarized as a decrease in the Amundsen and Bellingshausen Seas (60° – 120°W), with compensating increases in the western Ross Sea (150°E – 180°E), Weddell Sea and King Haakon VII Sea (50°W – 30°E). There is some seasonal dependence, and the Ross Sea is the only region

that has statistically-significant trends in all seasons. From this SIC pattern, we would expect a warming SAT trend in the region of the Antarctic Peninsula, and a cooling elsewhere, a pattern that is expressed by some of the reanalyses in Figure 1, but by no means all.

To quantify the level of agreement between SIC and SAT trends, we calculated correlations for each season and reanalysis amongst observed SIC, reanalysis SIC, and reanalysis SAT trends (Figure 3). Most of the reanalyses use boundary sea ice

conditions that match the passive microwave record reasonably-well, with trend pattern correlations consistently greater than 0.9 for many products, the best match being for ERA5 (Figure 3a). Both ERA-int and MERRA2 have a sea ice boundary condition that diverges somewhat from observations in late winter. The NCEP reanalyses (i.e. NCEP2 and CFSR), have a coupled, freely-evolving ocean-sea ice system, which explains the very low agreement with the satellite record compared to the products which use a prescribed sea ice condition.

Figure 3b shows the pattern correlation between reanalysis SAT trends and the trend of each reanalysis product's prescribed SIC trend. This serves as a test for the expectation that SAT and prescribed SIC trends should be internally-consistent, regardless of the actual SIC trend pattern. For most months and reanalyses this is indeed the case, with strong negative correlations for most of the year. NCEP2 is the exception and shows a positive correlation for much of the year, presumably because of its strong warming pattern in the sea ice zone (Figure 1) that is inconsistent with an increased sea ice cover in much

of the Antarctic domain. The sea ice-SAT trend relationship is strongest in the main sea ice growth season (March-July), consistent with previous model and observational studies showing that the sea ice-atmosphere relationship is stronger during the growth season (Raphael and Hobbs, 2014; Schroeter *et al*, 2017). The relationship is surprisingly weak in December and January, months combining high melt rates with relatively low sea ice cover. We explore this result in more detail in section 3.2, and note that for these months SIC trends may be a less reliable test for SAT trends.

Figure 3c shows the pattern correlations between observed SIC trends and reanalysis SAT, and therefore summarizes the evaluation of SAT in the reanalyses based on Antarctic sea ice trends. Other than for December-January when the SAT-SIC relationship is relatively weak, JRA55 and 20CRv3 have consistently the closest relationship between SAT and SIC trends, despite ERA5 having a closer correspondence between prescribed SIC and the satellite record (Figure 3a). ERA20C and ERA5 also have reasonable agreement for much of the year, but none of the ERA products have

any SAT-SIC trend correlation in December-January. ERA-int also has a notably weak SAT-SIC relationship in August, which seems to be largely due to the disagreement between the observed SIC trend and that of ERA-int's sea ice (Figure 3a), since the correlation between ERA-int SIC and SAT is strongly negative in August (Figure 3b).

Based on this analysis we would conclude that JRA55 and 20CRv3 have the best representations of long-term change over the polar Southern Ocean, under the assumption that SIC trends should be closely related to SAT trends. The summer SIC-SAT

relationship is much weaker in summer for all the products, but in the 3 ECMWF products analyzed here (ERA5, ERA-int and


ERA20C) this relationship completely disappears. We further tested the physical relationship between sea ice and air temperature by mapping the correlation coefficients between detrended, interannually-varying reanalysis SAT and SIC, for each calendar month (Supplemental Figures S25-S36). The results are similar to those for the trend pattern correlations, showing a strong negative correlation throughout the ice pack in most months, but which is weaker and more complex in

summer. In mid-winter the correlations are concentrated at the sea ice edge, where sea ice variability is greatest. The reanalyses with very high SIC have limited correlations within the ice pack, since sea ice concentrations > 0.9 must have limited variance and therefore weak covariance with SAT. In the next section, we explore the weak SAT-SIC relationship in December-January in more detail.

.

**3.2 Further exploration of summer SAT-sea ice relationship**

The correlations show that generally the relationship between SIC and SAT 30-year trends is weaker in December and January (Figure 3b). It is worthwhile considering the physical reasons for this weak summer relationship between SAT and sea ice, which are strongly coupled for the rest of the year. We note that December and January are months with very strong sea ice melt (Figure 2e), due to a surface heat flux from atmosphere to ocean. By contrast, for most of the year (March-October), the

heat flux is from ocean-to-atmosphere, driving the ocean cooling that allows sea ice to form, and for these months there is a positive relationship between the magnitude of cooling and the strength of the SAT-SIC correlation for the reanalyses (Figure 4), i.e. stronger cooling leads to a stronger negative SIC-Sat correlation. During these cold months, SAT is cooler where sea ice prevents a flux of heat from ocean-to-atmosphere and warmer over open water, which is the physical mechanism explain the negative correlation between SAT and SIC trends.

During warmer months when the mean flux is from atmosphere-to-ocean, this relationship breaks down or even becomes negative (Figure 4), indicating that surface heat flux is no longer a connection between ice and SAT variability. At a first pass we might still expect a relationship between the surface atmosphere and sea ice, since this heat flux is important for melting the sea ice. However, we suggest that 2 factors combine to break the SAT-SIC coupling. The first is that, from the perspective of the atmosphere (and in particular, an uncoupled atmosphere-only model which is the basis for most of the reanalyses), when

the flux is from atmosphere-to-ocean, the ocean is a passive sink of energy that is modulated by atmospheric processes, rather than an active driver of the surface atmosphere during periods of ocean-to-atmosphere energy transfer. We note that the CFSR – one of the only reanalyses to have a coupled rather than prescribed sea ice – has a stronger summer SAT-SIC correlation (Figure 3).

The second factor is the process of sea ice melt. Although the melt is largely driven by incoming solar radiation, there is in

fact relatively little melt on top of the sea ice because of the high albedo of snow-covered sea ice (Gordon, 1981; Drinkwater and Xiang, 2000). Instead, areas of open water such as leads absorb solar radiation, warming the ocean mixed layer and melting the ice pack from beneath (Stammerjohn *et al*, 2012). This means that impact of ice cover, which in summer mainly affects



the surface air temperature through reflecting solar radiation, is spatially diffused by the ocean and so reduces the direct spatial relationship between solar radiation, atmosphere, and sea ice.

A third and final factor, that is not immediately evident, is the nature of the summer ice pack, which although small in area comprises a higher proportion of thick, wind-compacted sea ice at the coastline than other months, since this is the ice most likely to survive the spring melt. This thick, compacted sea ice is relatively insensitive the atmospheric warming (Enomoto and Ohmura, 1990; Massom *et al*, 2008), and so the sea ice-atmosphere relationship is also weak. As a result of these factors, we argue that a weak summer relationship between SIC and SAT is expected from the physical conditions in high summer.

However, this does not explain why the ERA products have apparently no SIC-SAT trend correlation in summer. Further analysis of the spatial distribution of SIC and SAT trends in Figure 5 reveals a local inconsistency between summer sea ice trends and the ERA SAT trends; this inconsistency is shown by hatching, which shows where a sea ice reduction is accompanied by a local cooling, or *vice versa*. The ERA products all show, to a greater or lesser degree of statistical-significance, a cooling over the Amundsen-Bellingshausen Seas (60ºW-150ºW ), a region with an intense and well-observed

loss of summer sea ice (e.g. Parkinson, 2019). None of the other products show a statistically-significant cooling in the same region. Whilst it has been hypothesized from model simulations have shown that a surface cooling may lead to a loss of sea ice, due to complex ocean-sea ice feedbacks (Zhang, 2007), in this particular region the sea ice loss has been robustly attributed to increasing poleward airflow, which both dynamically constrains the ice extent and advects warm air to the region (Holland and Kwok, 2012; Hosking *et al*, 2013; Raphael *et al*, 2016), and would be expected to drive warmer SAT. Furthermore, whilst

there are no direct measurements of SAT in this region there are a number of station observation records on the west Antarctic Peninsula, and reconstructions of continental surface temperature; these all indicate a warming trend over the west Antarctic landmass (Steig *et al*, 2009; Nicolas and Bromwich, 2014) in response to the same increase in warm northerly airflow that has reduced Amundsen-Bellingshausen sea ice cover. This continental warming is most clearly evident in JRA55 (Figure 5), but clearly does not seem to be consistent with a surface cooling over the adjacent ocean. We therefore consider that the ERA

products must be considered with some degree of caution, especially for studies of change in the West Antarctic region.

## 4. Conclusions

Using the known close relation between sea ice cover and surface air temperature in polar oceans, we use satellite-observed of Antarctic sea ice trends as an independent validation of reanalysis trends over the polar Southern Ocean. Based on this analysis,

we find several reanalysis products that reproduce reasonable surface air temperature trends, with JRA55 showing the consistently highest agreement with observed sea ice throughout the year.

We find that the relationship between surface air temperature and sea ice concentration is strong for most months of the year except mid-summer (i.e. December and January). These are the only months of the year when the polar Southern Ocean is a sink rather than a source of the net surface heat flux, and although much of the heat is used to melt sea ice, the heat is distributed

by ocean processes, and the direct spatial correlation between sea ice and air temperature is relatively weak.



Although all eight of the reanalysis products that we analyze here have a weaker air temperature -sea ice relationship in summer, the ECMWF reanalyses (ERA5, ERA-int and ERA20C) have no correlation in summer at all. These seems to be due to their representation of a surface cooling in the Amundsen and Bellingshausen Seas, which is not consistent with a robustly-observed local sea ice loss since the late 1970s, nor with independent reconstructions of land surface temperature, which show

a warming on the west Antarctic region adjacent to the Amundsen-Bellingshausen Seas.

## Acknowledgements

This work was supported in part by the Antarctic Climate and Ecosystems Cooperative Research Centre, and the Australian Research Council Antarctic Gateway Partnership. Data analysis and visualization was performed using the NCAR Command

Language (http://dx.doi.org/10.5065/D6WD3XH5). The authors would also like to acknowledge the help and resources of the NCAR Research Data Archive (https://rda.ucar.edu/), and reanalysis.org in acquiring and processing the data.

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

| | Description | Citation | Reanalysis period | Spatial Resolution (lat x lon) | Algorithm |
|---|---|---|---|---|---|
| NCEP2 | NCEP-DOE AMIP II Reanalysis | Kobayashi *et al* (2015) | 1979-present | 2.5º x2.5º | 3D-VAR |
| CFSR | NCEP Climate Forecast System Reanalysis | Saha *et al* (2010) | 1979-present | 0.5º x 0.5º | 3D-VAR |
| MERRA2 | Modern Era Retrospective Analysis for Research and Applications, version 2 | Gelaro *et al* (2017) | **1980**-present | 0.5º x 0.625º | 3D-VAR |
| 20CRv3 | National Oceanic and Atmospheric Administration - Cooperative Institute for Research in Environmental Sciences 20$^{th}$ Century Reanalysis version 3 | Slivinski *et al* (2019) | 1836-2015 | 1º x 1º | Ensemble Kalman Filter |
| ERA5 | European Centre for Medium Range Weather Forecasting (ECMWF) Reanalysis version 5 | Hersbach *et al* (2018) | 1979-present | 0.25º x 0.25º | 4D-VAR |
| ERA-20C | ECMWF 20$^{th}$ Century Reanalysis | Poli *et al* (2016) | 1900-**2010** | 0.25º x 0.25º | 4D-VAR |
| ERA-int | ECMWF Interim Reanalysis | Dee *et al* (2011) | 1979-2019 | 0.75º x 0.75º | 4D-VAR |
| JRA55 | Japanese 55-year Reanalysis | Kobayashi *et al* (2015) | 1958-present | 1.25º x 1.25º | 4D-VAR |

**Table 1: Summary of reanalysis products used in this study. Bold years in the reanalysis periods indicate the first and final year of the period for which output is available for all products.**


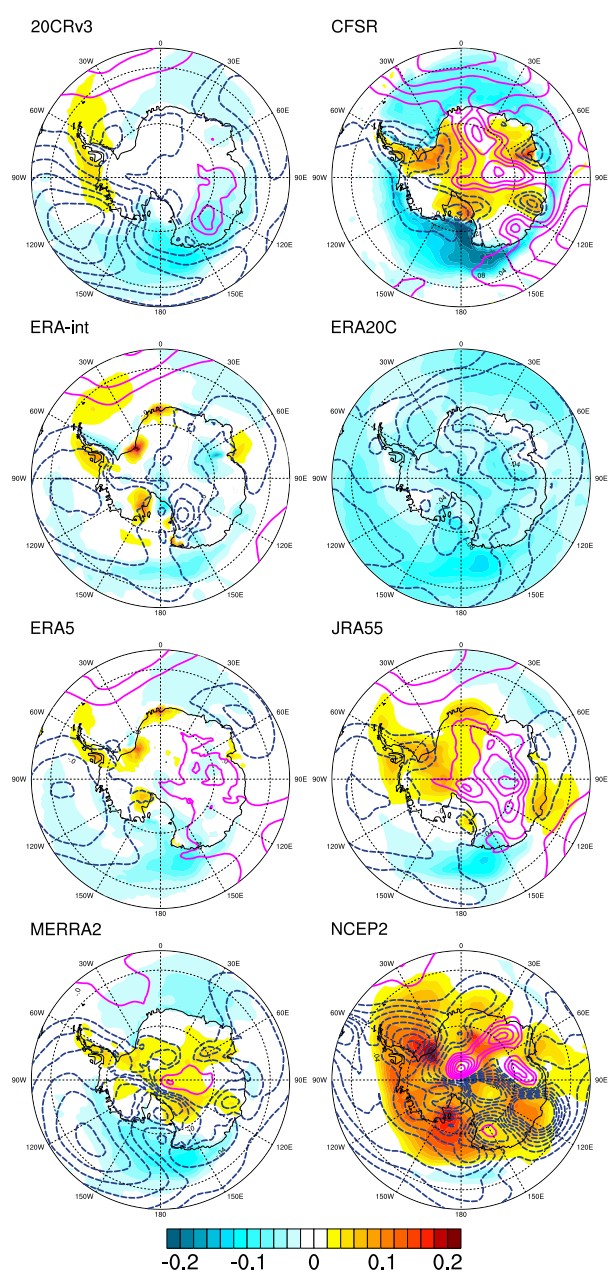

**Figure 1: 1980-2010 trends in annual-mean SAT (shading: ºC/year) and MSLP (contour lines: positive trends in blue, negative trends in magenta, with contour spacing = 2.5 Pa/year), for eight individual reanalyses (refer to Table 1 for details). Trend patterns for each month are shown in the Supplemental Material as Figures S1-S12.**




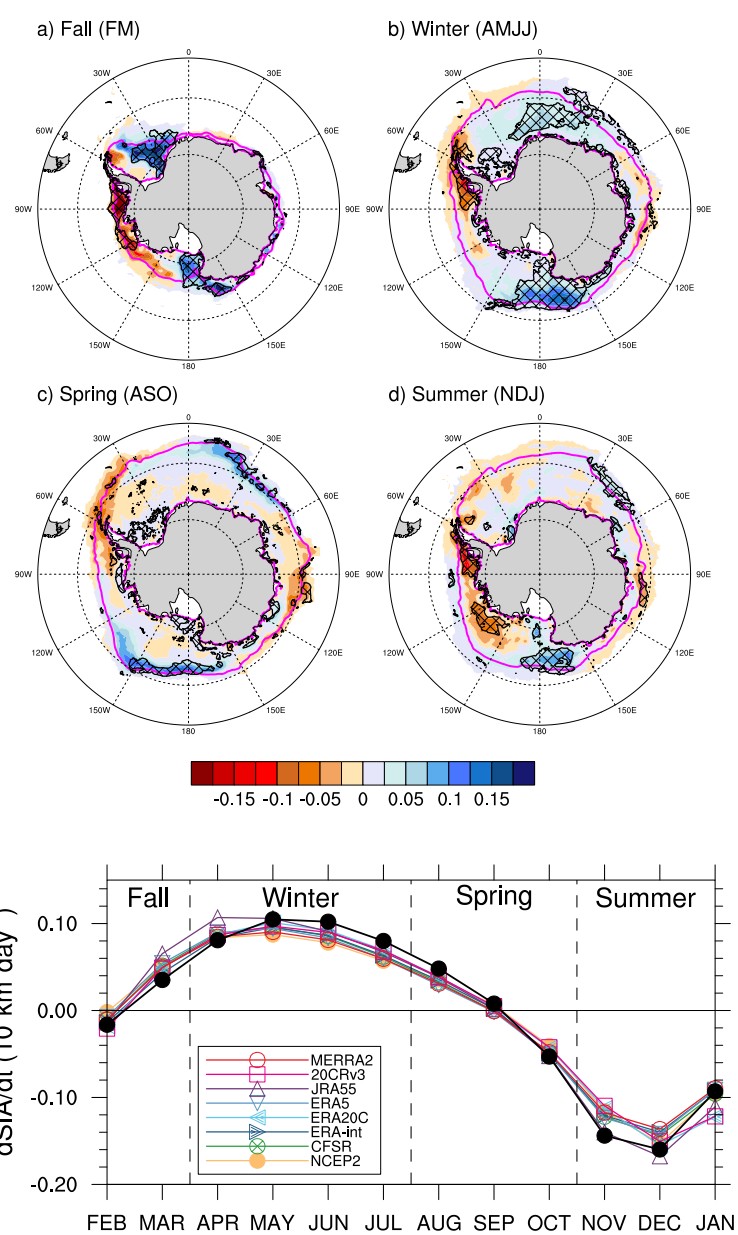

**Figure 2: Panels a-d show observed 1980-2010 Antarctic SIC trends by season (decade-1). Hatching indicates trends that are statistically-significant at the 0.05 significance level, and magenta lines show the climatological sea ice edge (defined by the 15% SIC isoline). Seasons are defined by total sea ice area (SIA) growth and melt (dSIA/dt), shown on panel e. Line colors in 2e follow the S-RIP standard.**





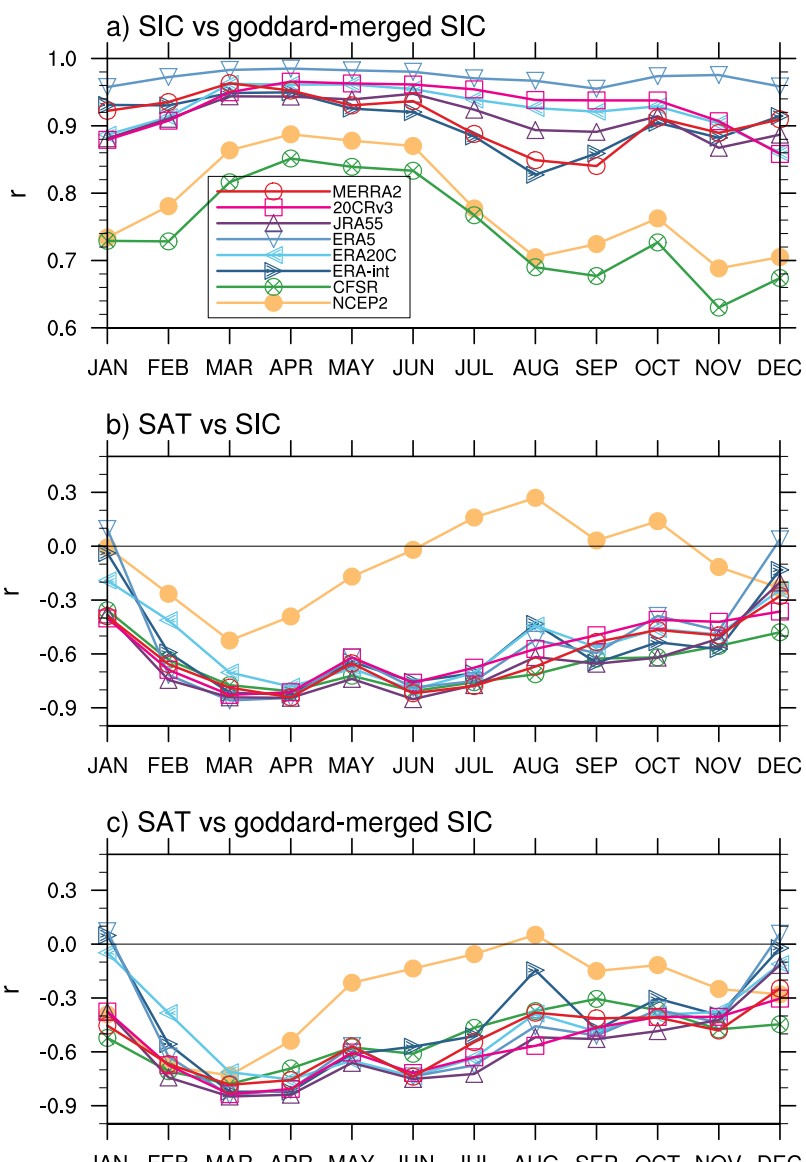

**Figure 3: Uncentered pattern correlations by month between 1980-2010 SIC and SAT trends: a) correlations between the observed SIC trend and each reanalysis' SIC; b) correlations between each reanalysis SAT trend that reanalysis' SIC trend; c) correlation between reanalysis SAT trend and observed SIC trend. Differences between each reanalysis SIC and the Goddard merged SIC product for each month are shown in the Supplemental Material as Figures S13-S24.**

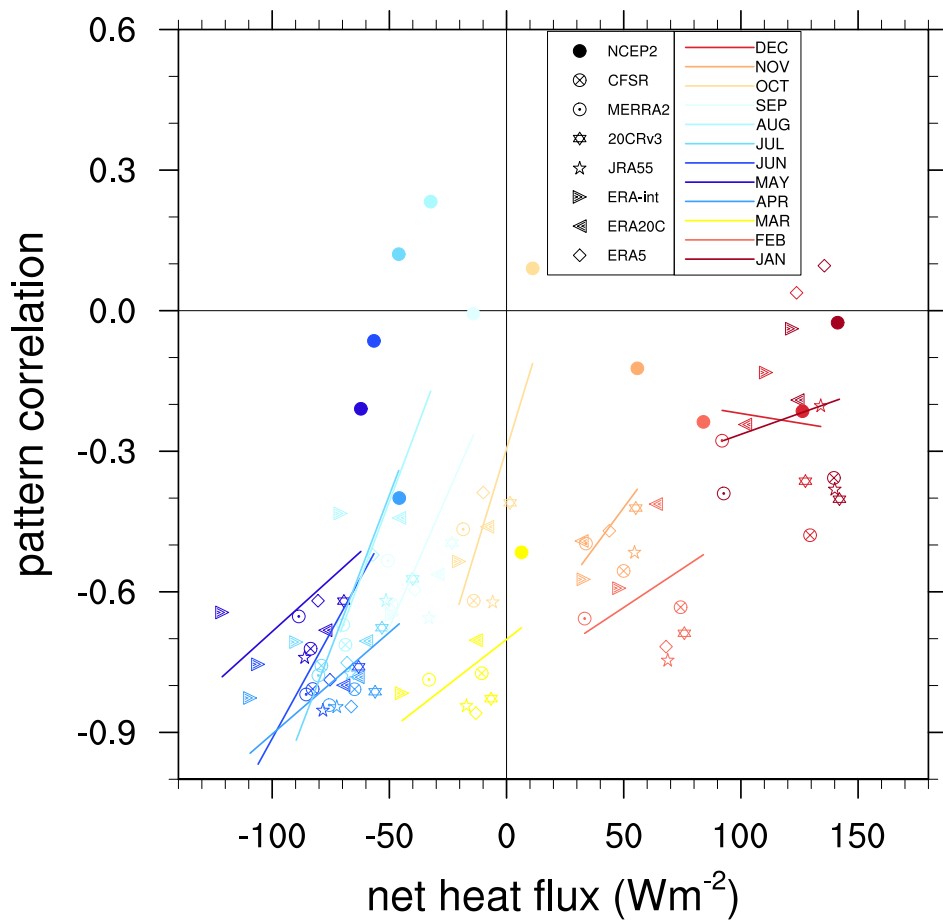


**Figure 4: Net heat flux averaged over sea ice zone (x-axis: Wm-2, where positive indicates net flux from atmosphere to ocean), against the pattern correlation between reanalysis SIC and Sat (i.e. values plotted in Figure 3b). Data points are colored by month with different markers for each reanalysis. Colored lines show the best-fit line for each month estimated by Ordinary Least Squares regression.**



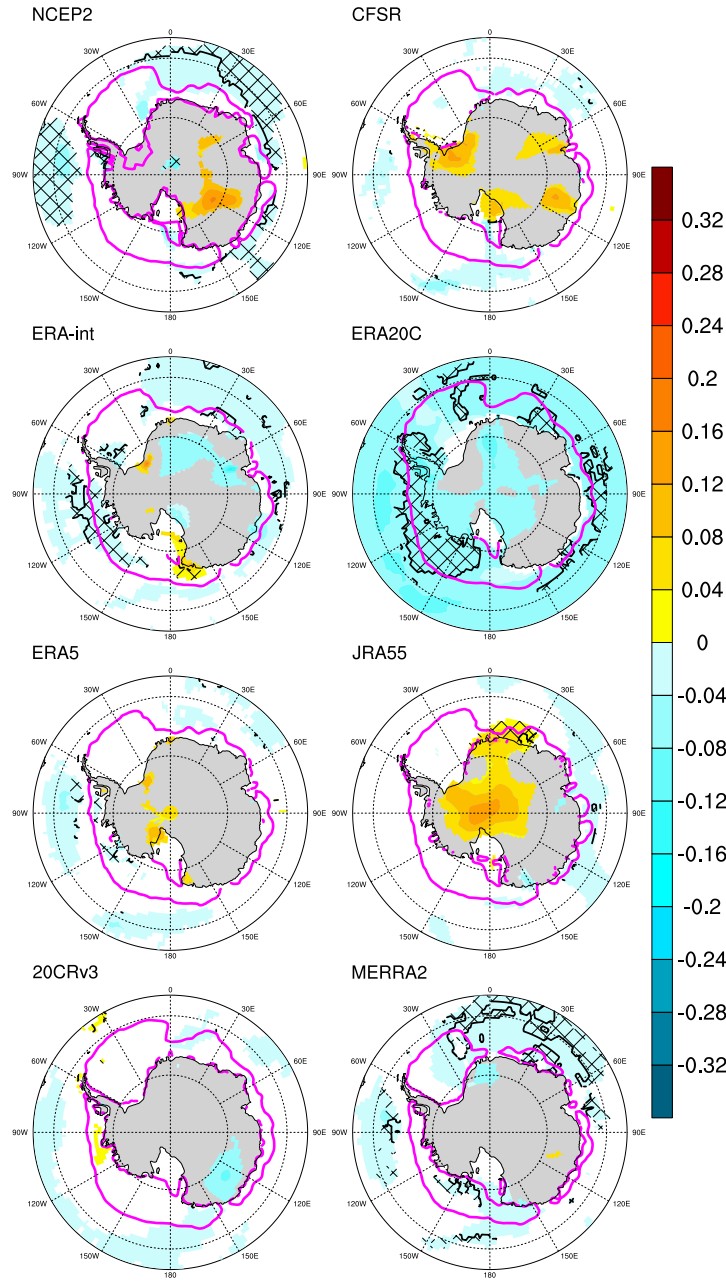

**Figure 5: Panels a)-h)1980-2010 December-January mean reanalysis SAT trends over the climatological sea ice zone (ºC/year); only trends that are statistically-significant at the 90% level are shown. Hatched regions show where the sign of statistically-significant reanalysis SIC and SAT trends are the same (i.e. the unexpected result of both a warming (cooling) and an increase (decrease) in sea ice cover). The magenta line shows climatological sea ice edge.**
