# Peer review of "Validation of reanalysis Southern Ocean atmosphere trends using sea ice data"

_Atmospheric Chemistry and Physics, 2020_

## Referee Comment (RC1) · Anonymous Referee #2 · 24 Aug 2020

The authors present an analysis of Southern Ocean surface air temperature (SAT) and sea-ice concentration (SIC) trends in 8 modern reanalysis products. The spatial patterns and magnitudes of SAT trends over the Southern Ocean are shown to vary greatly between reanalyses. They use the observed relationship between SAT and SIC, which is shown to be particularly strong during the sea-ice growth season, to validate reanalysis SAT trends. It is argued that the ERA reanalyses may give a less reliable representation of SAT trends, since they are less consistent with SIC changes. On the other hand, trends in JRA55 are proposed to be more reliable.

Constraining atmospheric trends over the Southern Ocean is a very important topic given the region's key role in global climate and the relative paucity of observations. I think that this paper plays a useful role in evaluating the reliability of SAT trends in

contemporary reanalysis, and as such will be of use to the wider community of users of reanalyses. The analysis is clearly presented and logical. I therefore recommend its publication in ACP, following just a few minor comments and technical corrections. I hope the authors find these to be useful.

Minor comments:

1. In discussing Fig 1 in Section 1 (showing SAT trends in the reanalyses), I think a note should be made of the fact that ERA20C is reliant on surface observations alone, of which there are very few in the Southern Ocean, so it is perhaps not surprising that it appears as an outlier.

2. It would be nice to see some measure of statistical significance in Fig 5 (for instance, as a gray shaded region). For 30 year time series I would guess that a lot of the correlations are not significant.

3. It is remarked that JRA55 is perhaps the best in terms of consistency between SAT and SIC trends. However, it is notable that it is quite an outlier in terms of the seasonal cycle of ice growth, having stronger growth rates in the fall/early winter than other reanalyses of observations (Fig 2e). I think the paper would benefit from some discussion of this – i.e. is JRA55 more reliable in terms of SAT trends, but perhaps less so in its seasonal cycle?

Technical corrections:

4. L62: particular -> particularly

5. Fig 2e is not labelled with an 'e'.

6. L151: 'Sat' -> 'SAT'

7. L181: This sentence doesn't make sense. I think 'have shown' needs to be removed.

---

## Referee Comment (RC2) · Anonymous Referee #1 · 1 Oct 2020

Firstly, I am really sorry that this review took so long for me to do.

This is an interesting and useful paper, in which the authors use the relationship between surface air temperature (SAT) and sea ice concentration (SIC) to evaluate a selection of reanalyses over the Antarctic. The analysis is simple (which I like) and effective, and the paper well written, and provides useful information for potential users of reanalysis data. I therefore recommend acceptance subject to minor revisions.

- Figure 3: marking a horizontal line with the $p < 0.05$ significance would aid interpretation. Also in the paragraph that discusses these results, comment on the significance of the correlations.

- Line 133. You conclude that JRA55 and 20CRv3 have the best representations of

long-term change over the Southern Ocean. I would clarify the phrasing here that they have the best representation over the study period. As JRA55 goes back to 1958, and 20CRv3 back to 1836, this could send the message to an uncautious reader that this comment applies to earlier periods. Given the lower amounts of data going into both reanalyses in earlier period, we do not know whether this conclusion holds for earlier periods (and you obviously can't test it due to lack of earlier sea-ice data).

- Line 134 and 158, write numbers <10 in full 'Three' and 'Two'.

- Table 1: I think the '1980' in the reanalysis period box for MERRA 2 may be in bold

- Figure 1 is labelled as Figure 2.

- In the actual Figure 2, panel e needs clearer description. Define what 'S-RIP' is, and also are the black dots/line observations?

---

## Author Comment (AC1) · 5 Oct 2020

We thank the Reviewer for their positive and helpful suggestions. We respond to their comments below

-Figure 3: marking a horizontal line with the $p < 0.05$ significance would aid interpretation. Also in the paragraph that discusses these results, comment on the significance of the correlations.

Author Response

Estimating a statistical significance for the pattern correlations in Figure 3 is very difficult, since it would require a calculation of spatial decorrelation length scales (zonal and meridional) in sea ice and SAT, in order to accurately estimate the degrees-of-

freedom (I.e. number of independent spatial pints in the trend patterns); this is a much more complex task than estimating significance of temporal correlations (as we have done for Figure 5).

Whilst we agree that it would be a nice addition to have such a significance estimate, it would greatly increase the complexity of the Methods (as the Reviewer notes, the intuitive simplicity of the Method is a strength of the paper), and it is our opinion that this would not alter the conclusions.

-Line 133. You conclude that JRA55 and 20CRv3 have the best representations of long-term change over the Southern Ocean. I would clarify the phrasing here that they have the best representation over the study period. As JRA55 goes back to 1958, and 20CRv3 back to 1836, this could send the message to an uncautious reader that this comment applies to earlier periods. Given the lower amounts of data going into both reanalyses in earlier period, we do not know whether this conclusion holds for earlier periods (and you obviously can't test it due to lack of earlier sea-ice data).

Author Response

This is an important caveat, and we have amended the sentence to read as follows:

'Based on this analysis we would conclude that JRA55 and 20CRv3 have the best representations of change since the late 1970s over the polar Southern Ocean, under the assumption that SIC trends should be closely related to SAT trends. We note that this may not hold true for earlier periods which are unconstrained by satellite retrievals, and for which we do not have reliable sea ice observations.'

-Line 134 and 158, write numbers <10 in full 'Three' and 'Two'.

We have edited the manuscript accordingly

- Table 1: I think the '1980' in the reanalysis period box for MERRA 2 may be in bold

Author Response

[Figure]

Bold years indicated the first and last year for which data is avaliable from all the re-analyses (1980 constrained by MERRA2; 2010 constrained by ERA20C. However, on consideration we feel the bold type does not add to the information, and have removed the bold type, and amended the Table caption accordingly.

- Figure 1 is labelled as Figure 2.

Author Response

Typo corrected

- In the actual Figure 2, panel e needs clearer description. Define what 'S-RIP' is, and also are the black dots/line observations?

Author Response

Figure 2e legend amended to include 'Observations' for black lines/markers. S-RIP abbreviation is now expanded to Figure 2 caption

---

## Author Comment (AC2) · 5 Oct 2020

We thank the Reviewer for their positive and helpful suggestions. We respond to their comments below

1. In discussing Fig 1 in Section 1 (showing SAT trends in the reanalyses), I think a note should be made of the fact that ERA20C is reliant on surface observations alone, of which there are very few in the Southern Ocean, so it is perhaps not surprising that it appears as an outlier.

Author Response

We have added the following sentence to this section:

[Figure]

'Much of this spread is due to differences in the forecast model and assimilation technique, but it should be noted that some of the products (ERA20C and 20CRv3) are not constrained by satellite data in order to give a consistent product over long historical periods; this is a major limitation in the remote Antarctic region.'

2. It would be nice to see some measure of statistical significance in Fig 5 (for instance, as a gray shaded region). For 30 year time series I would guess that a lot of the correlations are not significant.

Author Response

As noted in the Figure 5 caption, only SAT trends that are significant at the 90% confidence level are plotted, and only areas where both the SAT and SIC trends are statistically-significant at the 90% level (and the SAT and SIC trends are inconsistent) are hatched.

3. It is remarked that JRA55 is perhaps the best in terms of consistency between SAT and SIC trends. However, it is notable that it is quite an outlier in terms of the seasonal cycle of ice growth, having stronger growth rates in the fall/early winter than other reanalyses of observations (Fig 2e). I think the paper would benefit from some discussion of this – i.e. is JRA55 more reliable in terms of SAT trends, but perhaps less so in its seasonal cycle?

Author Response

We have added the following sentences to our summary fo Figure 3:

'We note that this may not hold true for earlier periods which are unconstrained by satellite retrievals, and for which we do not have reliable sea ice observations. An interesting point to note is that although JRA55 performs well with respect to this metric, it has the strongest sea ice bias in March and April. This raises the question of whether the mean state is a good indicator of performance in respect of variability or trends.'

4. L62: particular -> particularly

Author Response

Typo corrected

5. Fig 2e is not labelled with an 'e'.

Author Response

Figure caption added to panel 2e

6. L151: 'Sat' -> 'SAT'

Author Response

Typo corrected

7. L181: This sentence doesn't make sense. I think 'have shown' needs to be removed

Author Response

Correct, we have corrected this sentence which now reads:

'Whilst it has been hypothesized from model simulations that due to complex ocean-sea ice feedbacks, a surface cooling may lead to a loss of sea ice (Zhang, 2007), we note that in this particular region the sea ice loss has been robustly attributed to increasing poleward airflow, which both dynamically constrains the ice extent and advects warm air to the region (Holland and Kwok, 2012; Hosking et al, 2013; Raphael et al, 2016), and would be expected to drive warmer SAT.'